# Unifying Heterogeneous Multi-Modal Remote Sensing Detection Via Language-Pivoted Pretraining

**Yuxuan Li**[1]  **Yuming Chen**[1]  **Yunheng Li**[1]  **Ming-Ming Cheng**[1 2]  **Xiang Li**[† 2 1]  **Jian Yang**[† 1 3]

## Abstract

Heterogeneous multi-modal remote sensing object detection aims to accurately detect objects from diverse sensors (e.g., RGB, SAR, Infrared). Existing approaches largely adopt a late alignment paradigm, in which modality alignment and task-specific optimization are entangled during downstream fine-tuning. This tight coupling complicates optimization and often results in unstable training and suboptimal generalization. To address these limitations, we propose **BabelRS**, a unified language-pivoted pretraining framework that explicitly decouples modality alignment from downstream task learning. BabelRS comprises two key components: Concept-Shared Instruction Aligning (CSIA) and Layerwise Visual-Semantic Annealing (LVSA). CSIA aligns each sensor modality to a shared set of linguistic concepts, using language as a semantic pivot to bridge heterogeneous visual representations. To further mitigate the granularity mismatch between high-level language representations and dense detection objectives, LVSA progressively aggregates multi-scale visual features to provide fine-grained semantic guidance. Extensive experiments demonstrate that BabelRS stabilizes training and consistently outperforms state-of-the-art methods without bells and whistles. Code: github.com/zcablii/SM3Det.

## 1. Introduction

Heterogeneous multi-modal remote sensing (RS) object detection (Li et al., 2025a; Yang et al., 2021; Dai et al., 2024a; Li et al., 2024a; Dai et al., 2024b) relies on data acquired

[1]PCA Lab & VCIP, CS, Nankai University. [2]NKIARI, Shenzhen Futian. [3]PCA Lab, School of Intelligence Science and Technology, Nanjing University.. Correspondence to: Xiang Li and Jian Yang <{xiang.li.implus;csjyang}@nankai.edu.cn>.

*Proceedings of the 43$^{rd}$ International Conference on Machine Learning*, Seoul, South Korea. PMLR 306, 2026. Copyright 2026 by the author(s).

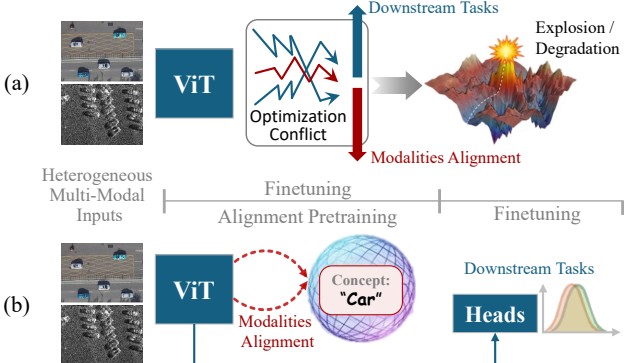

*Figure 1.* Conceptual comparison between (a) late alignment and (b) early, language-pivoted alignment paradigms for heterogeneous multi-modal remote sensing detection. Late alignment (a) entangles modality alignment with task optimization during fine-tuning, leading to gradient conflicts and unstable training. BabelRS (b) decouples these objectives via early semantic alignment, resulting in improved optimization stability and generalization.

from diverse sensors operating under fundamentally different imaging principles. These differences give rise to distinct modalities whose representations vary substantially in structure and semantics. Recent research efforts (Li et al., 2026) aim to develop a unified model capable of processing all such modalities within a single model. Despite encouraging progress, current state-of-the-art models still encounter an intrinsic limitation in their underlying learning paradigm.

Existing approaches, such as SM3Det (Li et al., 2026), typically follow a "late alignment" strategy as shown in Figure 1. They initialize models using generic unimodal backbones, then attempt to align heterogeneous feature spaces while simultaneously optimizing detection objectives during fine-tuning. This joint optimization is particularly challenging when modalities exhibit intrinsically different physical characteristics, such as the contrast between scattering mechanisms in SAR imagery and reflectance-based signals in RGB images. Both our empirical observations and theoretical analysis (in the Appendix) suggest this paradigm causes unstable optimization dynamics. The issue becomes more pronounced as model capacity increases, for example when employing large-scale backbones such as ViT-Large or incorporating heavy dynamic components like Mixture-

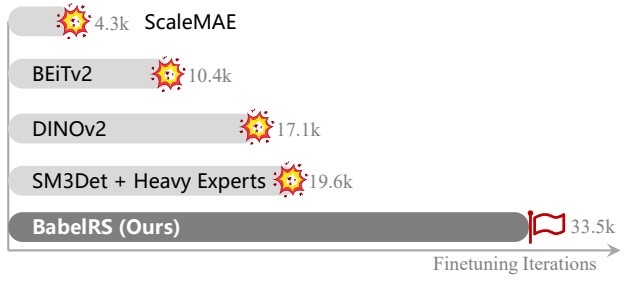

*Figure 2.* Automatic Mixed Precision fine-tuning stability on SOI-Det dataset. Many existing models experience gradient explosion before completion, whereas BabelRS remains stable throughout fine-tuning.

of-Experts (Jacobs et al., 1991). In practice, as illustrated in Figure 2, this instability often manifests as gradient explosions or numerical failures, including NaN values.

Could these issues be alleviated by separating modality alignment from task learning altogether? Large-scale multimodal pretraining offers a natural direction, as it enables alignment to be learned before downstream optimization. However, existing pretraining frameworks (Cong et al., 2022; Guo et al., 2023; Bachmann et al., 2022) rely on strong assumptions about data availability. Most require spatially aligned multi-modal image pairs, such as paired optical–SAR images used in SkySense (Guo et al., 2023). In realistic RS scenarios, especially for more complex configurations involving RGB, SAR, and infrared sensors, collecting such spatially homogeneous data at scale is often infeasible. This scarcity of such data therefore constitutes a fundamental bottleneck for unified multimodal learning.

To overcome this constraint, we introduce **BabelRS**, which reframes the RS multi-modal learning paradigm from "late alignment" to "early alignment" via language-pivoted pretraining. While pixel-level correspondence across modalities is difficult to obtain, semantic correspondence is naturally available through language. For example, a SAR image of a car and an RGB image of the same category object both correspond to the linguistic concept "Car". Building on this fact, BabelRS leverages language as a semantic anchor to align heterogeneous modalities before task-specific training.

Specifically, we introduce Concept-Shared Instruction Aligning, which treats the embedding space of a large language model as a shared semantic reference. During pretraining, images from different modalities are aligned to the same linguistic concepts through instruction-following objectives, guiding the unified visual encoder to produce consistent semantic representations in language space. Consequently, images depicting the same object are mapped to similar features regardless of whether they originate from RGB, SAR, or infrared sensors.

Yet does semantic alignment alone suffice for dense detec-

tion pretraining? Existing vision-language models, such as CLIP (Radford et al., 2021) and InternVL (Gao et al., 2024), typically align language only with the final layer of a Vision Transformer (ViT) (Dosovitskiy et al., 2020). While effective for global semantic reasoning, this design introduces a granularity mismatch when applied to object detection, which demands multi-scale and spatially resolved features. To address this limitation, we introduce Layerwise Visual-Semantic Annealing, which progressively integrates intermediate ViT representations into the language-aligned space. It allows the pretraining process preserve the joint calibration of low- and high-level visual features, enabling precise localization while retaining strong semantic guidance from LLMs.

Our contributions are summarized as follows:

1. We identify optimization conflicts in late alignment as a key source of training instability in heterogeneous multi-modal remote sensing detection.

2. We propose **BabelRS**, a language-pivoted pretraining framework that achieves RS cross-modal representation alignment, through Concept-Shared Instruction Aligning and Layerwise Visual-Semantic Annealing.

3. Experiments demonstrate that BabelRS yields stable optimization and achieves new SOTA performance.

## 2. Related Work

### 2.1. Multi-Modal Remote Sensing Object Detection

The rapid development of remote sensing platforms has led to the widespread availability of multi-modal data, including optical imagery, Synthetic Aperture Radar (SAR), and infrared observations. Each modality captures different physical properties of the Earth's surface. For instance, optical sensors provide rich textural detail but depend on illumination, while SAR offers all-weather capability but lacks color information. These complementary characteristics have motivated extensive research into multi-modal object detection, with the goal of improving robustness and generalization beyond what single-modality systems can achieve.

#### 2.1.1. PAIRED MULTI-MODAL FUSION (SPATIALLY HOMOGENEOUS)

Early work on multi-modal remote sensing detection largely assumes access to spatially aligned image pairs (Sun et al., 2022; Guo et al., 2023). Under this setting, feature-level fusion becomes the dominant paradigm, as it enables flexible integration of heterogeneous information at different abstraction levels. Several studies introduce adaptive fusion strategies that dynamically weight modalities based

on scene conditions. Illumination-aware models (Li et al., 2019; Guan et al., 2019) adjust RGB and infrared contributions according to lighting cues. Fusion CSPNet (Wolpert et al., 2020) focuses on improving detection for small or occluded pedestrians through specialized fusion blocks. To mitigate modality dominance, Differential Modality Aware Fusion (Zhou et al., 2020) explicitly calibrates modality-specific feature importance. Spatial misalignment presents another major obstacle in paired fusion. AR-CNN (Zhang et al., 2021) proposes region-level alignment to handle positional shifts between modalities, while TSFADet (Yuan et al., 2022) introduces explicit translation modules to align misregistered feature maps. More recent architectures, such as C2Former (Yuan & Wei, 2024), leverage cross-attention and adaptive sampling to improve fusion precision under miscalibration. DMM (Zhou et al., 2025b) further explores efficient cross-modal fusion using a Mamba-based model. Despite strong performance, these methods fundamentally rely on spatially homogeneous data. Their assumptions break down when paired observations are unavailable or unreliable, which is often the case in real-world remote sensing scenarios.

### 2.1.2. UNIFIED MULTI-MODAL LEARNING (SPATIALLY HETEROGENEOUS)

To relax the requirement for paired inputs, recent research has explored unified models capable of processing spatially heterogeneous data. The objective is to train a single detector that generalizes across modalities, allowing inference on RGB, SAR, or infrared images independently. SM3Det (Li et al., 2026) represents an early attempt in this direction. While effective, it largely follows a late alignment strategy, where modality alignment is attempted only during supervised fine-tuning. Backbones are typically initialized from generic RGB pretraining, and alignment emerges implicitly through the detection objective. This formulation introduces an optimization dilemma: the model must simultaneously reconcile heterogeneous feature distributions and learn task-specific representations. As modality divergence increases, this coupling often leads to unstable training and limited generalization. In contrast, BabelRS explicitly decouples these objectives. By shifting alignment to a dedicated pretraining stage, the proposed framework alleviates the optimization conflict inherent in late alignment paradigms.

### 2.2. Multi-Modal Alignment

Vision–language alignment has demonstrated remarkable success in general domains. CLIP (Radford et al., 2021) establishes a shared embedding space through contrastive learning, enabling robust zero-shot transfer. Image-Bind (Girdhar et al., 2023) extends this idea by treating images as a binding interface to align diverse modalities. In contrast, LanguageBind (Zhu et al., 2023) places language

at the center, aligning modalities such as infrared and video directly to a frozen LLM. Domain-specific extensions further illustrate the flexibility of language-pivoted alignment. MolBind (Xiao et al., 2024) aligns language with molecular and protein representations, while Babel (Dai et al., 2025) proposes an expandable framework for scalable sensor integration. UNIALIGN (Zhou et al., 2025a) introduced a unified architecture to scale alignment across massive multimodal datasets effectively. Most of these methods emphasize global semantic alignment or rely on paired data (e.g., RGB-depth), which limits their applicability to heterogeneous remote sensing modalities. BabelRS differs in two key aspects. It achieves implicit cross-modal alignment without multi-sensor paired data, and it explicitly addresses the granularity mismatch between language semantics and dense detection through a Layerwise Visual-Semantic Annealing mechanism.

## 3. Remote Sensing Visual Encoder Pretraining

Early advances in remote sensing pre-training relied on supervised learning with curated datasets, such as LSKNet (Li et al., 2023), SAMRS (Wang et al., 2023a) and MSFA (Li et al., 2024c). While effective, these approaches depend heavily on annotated data and struggle to scale across modalities. Masked Image Modeling (MIM) (He et al., 2022) has since become a standard approach for leveraging large volumes of unlabeled remote sensing imagery. SatMAE (Cong et al., 2022) adapts MAE to capture temporal dynamics, while RingMo (Sun et al., 2023) and ScaleMAE (Reed et al., 2023) address dense object distributions and resolution variability. Contrastive learning has also gained traction, primarily through image–text alignment for zero-shot learning (Zhang et al., 2024b; Liu et al., 2024) and image–image discrimination (Guo et al., 2023; Zhang et al., 2025). Hybrid frameworks, such as CMID (Muhtar et al., 2023) and GFM (Mendieta et al., 2023), combine MIM and contrastive objectives to improve robustness. More recently, foundation models for remote sensing have shifted toward large-scale unification. OFA-Net (Xiong et al., 2024) emphasizes architectural versatility across tasks, while msGFM (Han et al., 2024) targets multi-sensor alignment. SkySense (Guo et al., 2023) and SkySense V2 (Zhang et al., 2025) scale this paradigm using massive paired optical–SAR datasets. ViTP (Li et al., 2025b) introduces instruction-following objectives to enforce perception capabilities into the backbone from the high-level understanding supervision. Despite these advances, existing pre-training strategies remain constrained to single-modal data or strictly paired multi-modal inputs. To date, no framework has focused on pretraining visual encoders by aligning cross-modal semantics using spatially heterogeneous data. BabelRS addresses this gap by exploiting language as a universal semantic anchor.

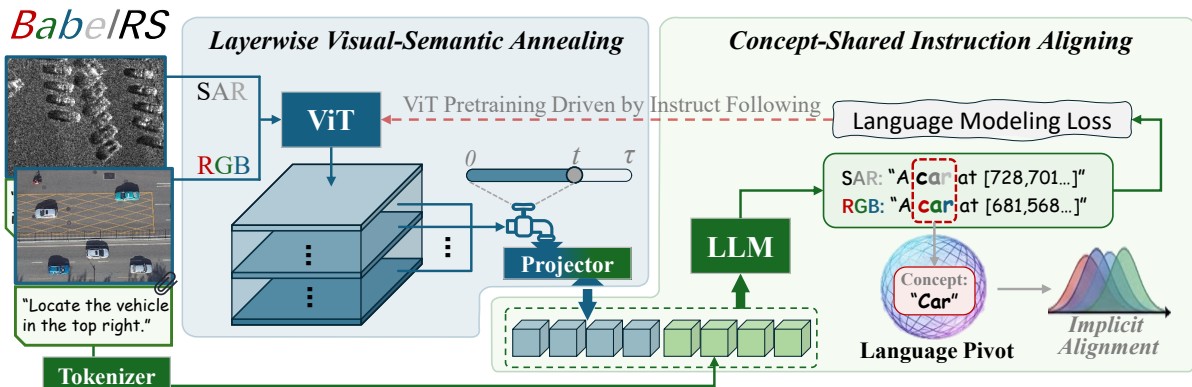

*Figure 3.* Overview of the BabelRS framework. BabelRS consists of two key components: Concept-Shared Instruction Aligning, which aligns heterogeneous remote sensing modalities into a shared linguistic semantic space using instruction-following objectives, and Layerwise Visual-Semantic Annealing, which progressively integrates multi-scale visual features into the language-aligned representation to support dense object detection.

# 4. Method

Existing unified detection frameworks for spatially heterogeneous remote sensing data typically entangle modality alignment with task learning. BabelRS departs from this design by introducing a language-pivoted pretraining strategy that separates these objectives. As illustrated in Figure 3, the framework consists of Concept-Shared Instruction Aligning and Layerwise Visual-Semantic Annealing.

Concept-Shared Instruction Aligning maps visual representations from different remote sensing modalities into a shared linguistic space, enabling implicit cross-modal alignment without requiring spatially heterogeneous data. Layerwise Visual-Semantic Annealing then bridges the granularity gap between language-level semantics and dense detection requirements by progressively integrating multi-scale visual features.

## 4.1. Concept-Shared Instruction Aligning

Let $\mathcal{M}$ denote a set of $K$ remote sensing modalities (e.g., RGB, SAR, infrared):

$$\mathcal{M} = \{m_1, m_2, \ldots, m_K\}. \quad (1)$$

Unlike prior approaches that necessitate paired samples $(x^{rgb}, x^{sar})$, we consider a collection of disjoint multi-modal remote sensing datasets $\mathcal{D}$:

$$\mathcal{D} = \{\mathcal{D}^{m_1}, \mathcal{D}^{m_2}, \ldots, \mathcal{D}^{m_K}\}, \quad (2)$$

where each dataset $\mathcal{D}^{m_k} = \{(x_i^{m_k}, q_i^{m_k}, r_i^{m_k})\}$ consists of image-text pairs. In this context, $q_i$ represents a natural-language question or instruction associated with image $x_i$ and $r_i$ is the corresponding text answer or response. These instruction-response data describe objects, spatial relations, or scene attributes visible in the image.

The central hypothesis of this work is that while pixel distributions $P(x^m)$ and imaging mechanisms vary significantly across modalities, their semantic interpretations can be expressed through shared linguistic concepts. Specifically, we assume the existence of modality-specific functions that map observations to a common linguistic concept $C$:

$$f(x^{rgb}) \rightarrow C, \quad g(x^{sar}) \rightarrow C. \quad (3)$$

Under this formulation, $f(x^{rgb})$ and $g(x^{sar})$ become implicitly aligned within the induced feature space. We formalize this intuition by using a pretrained Large Language Model $\Phi$ as a semantic pivot. For each modality, a modality-shared vision encoder $E_{\mathcal{M}}$ extracts visual features, which are projected into the input embedding space of $\Phi$. Given an image $x$ from any modality within $\mathcal{M}$ and its associated text $\{q, r\}$, we optimize a causal language modeling objective:

$$\mathcal{L}_{\text{align}} = - \sum_{j=1}^{|r|} \log P_{\Phi}(r_j \mid q, r_{<j}, E_{\mathcal{M}}(x)). \quad (4)$$

By forcing heterogeneous modalities to produce consistent linguistic descriptions, the vision encoder is guided toward a shared semantic manifold. In practice, we adopt an instruction-following (Liu et al., 2023) paradigm by concatenating visual tokens with textual tokens corresponding to the instruction and response. The language modeling loss is applied only to the response tokens. This design not only projects visual representations into a unified linguistic space but also leverages the compositional and reasoning demands of instruction-following tasks to encourage the vision encoder to capture informative, semantically grounded features. Importantly, this alignment process is performed independently of downstream detection objectives, which leads to more stable optimization and improved generalization across modalities.

## 4.2. Layerwise Visual-Semantic Annealing Mechanism

While semantic alignment provides a strong foundation, dense object detection requires spatially resolved, multi-scale features extracted from multiple depths of the visual backbone. Most vision–language models align language only with the final ViT layer, which captures global semantics. Directly aggregating all intermediate layers can disrupt pre-trained feature distributions and introduce instability. To address these issues, we propose **Layerwise Visual-Semantic Annealing** (LVSA) mechanism. This mechanism effectively aggregates multi-scale information while mitigating sudden distribution shifts, thereby maintaining the integrity of the pretrained backbone. Let $\mathcal{V}$ represent the set of feature maps from the $L$ blocks of the ViT encoder:

$$\mathcal{V} = \{F_l\}_{l=1}^L, \quad F_l \in \mathbb{R}^{H \times W \times C}. \quad (5)$$

We define a subset of selected layers for fusion as $\mathcal{S} \subseteq \{1, \ldots, L\}$, where the final layer $L \in \mathcal{S}$. To transition smoothly from single-scale to multi-scale representations, we introduce a time-dependent fusion coefficient $\alpha(t)$, governed by the training step $t$ and a annealing duration $\tau$, to control the contribution of multi-scale features:

$$\alpha(t) = \min\left(\frac{t}{\tau}, 1\right). \quad (6)$$

The fused feature representation, $\tilde{F}$, is calculated as a dynamic interpolation between the final layer $F_L$ and the mean of all selected features:

$$\tilde{F} = (1 - \alpha(t))F_L + \alpha(t)\left(\frac{1}{|\mathcal{S}|}\sum_{l \in \mathcal{S}} F_l\right). \quad (7)$$

At early stages, the model relies primarily on the final layer, preserving the original pre-trained distribution. As training progresses ($t \to \tau$), lower-level features are gradually incorporated, enabling precise localization while avoiding abrupt distribution shifts. By explicitly pretraining intermediate layers under semantic guidance, multi-scale features required by downstream detectors are both spatially informative and semantically consistent. LVSA happens simultaneously with Concept-Shared Instruction Aligning during the pretraining stage.

## 4.3. Task-Specific Fine-tuning

After pretraining, the aligned encoder is fine-tuned for heterogeneous multi-modal object detection. Unlike prior frameworks that introduce additional alignment modules during fine-tuning, BabelRS adopts a streamlined design. A shared backbone is combined with modality-specific detection heads, and training proceeds using a random sampling strategy across datasets. The total loss is defined as the sum of task-specific losses $n$,

$$Loss_{\text{total}} = \sum_n Loss_n. \quad (8)$$

By removing auxiliary alignment objectives, fine-tuning focuses entirely on detection performance.

## 4.4. Harmonic Modality mAP (H-mAP)

Multi-modal remote sensing datasets often exhibit significant category imbalance across modalities. Let $\mathcal{C}_m$ represent the categories specific to modality $m$. In practice, $|\mathcal{C}_{RGB}|$ is typically much larger than $|\mathcal{C}_{SAR}|$ or $|\mathcal{C}_{IR}|$ in open-source datasets such as SOI-Det (Li et al., 2026) dataset. Standard evaluation using Global Mean Average Precision ($mAP$) calculates the mean over the union of all categories:

$$\mathcal{C}_{total} = \bigcup_{m \in \mathcal{M}} \mathcal{C}_m \ , \quad (9)$$

$$mAP = \frac{1}{|\mathcal{C}_{total}|} \sum_{c \in \mathcal{C}_{total}} AP_c \ . \quad (10)$$

Due to the disparity in category counts, $mAP$ is implicitly biased toward the RGB modality. A model may achieve a high global score while failing to generalize to SAR or IR domains. To address this, we propose the Harmonic Modality mAP (H-mAP) to better reflect cross-modal reliability. We first compute the modality-specific mAP:

$$mAP_m = \frac{1}{|\mathcal{C}_m|} \sum_{c \in \mathcal{C}_m} AP_c, \quad \forall m \in \mathcal{M} \quad (11)$$

Then $H\text{-}mAP$ is defined using the harmonic mean:

$$H\text{-}mAP = \frac{|\mathcal{M}|}{\sum_{m \in \mathcal{M}} \frac{1}{mAP_m}} \quad (12)$$

The choice of the Harmonic Mean is motivated by its sensitivity to outliers and minimum values. Unlike the Arithmetic Mean, which allows high performance in one domain to compensate for failure in another, the Harmonic Mean enforces a "weakest link" constraint. Mathematically, if the performance in any single modality $m$ approaches zero ($mAP_m \to 0$), the overall score converges to zero ($H\text{-}mAP \to 0$), regardless of the performance in other modalities. This formulation penalizes modality-specific failures and rewards balanced performance, ensuring that improvements are not driven by a single dominant modality.

## 5. Experiments

### 5.1. Pretraining Dataset

The composition of the pretraining corpus is summarized in the Table 1. We integrate diverse large-scale remote sensing vision–language datasets to ensure broad semantic coverage across modalities and tasks. Million-AID (Long et al., 2021), LevirCC (Liu et al., 2022), VHM (Pang et al., 2025),

RSVQA (Lobry et al., 2020), and FIT_RS (Luo et al., 2024) provide large-scale visual instruction data for scene understanding, object recognition, and attribute reasoning, while GAIA (Zavras et al., 2025) further extends coverage with meteorological and multispectral imagery. Together, these datasets span diverse regions, resolutions, and semantic concepts. To enhance SAR-specific alignment, we incorporate SARLang (Wei et al., 2025), a large-scale SAR-centric VQA dataset emphasizing location-aware queries and fine-grained target semantics. For infrared imagery, we use MMRS-1M (Zhang et al., 2024a), retaining only infrared samples to introduce thermal signatures. GeoChat (Kuckreja et al., 2024), DIOR-RSVG (Zhan et al., 2023), and VRS-Bench (Li et al., 2024b) further provide visual grounding annotations linking language to spatial regions and object extents. All datasets are reprocessed and standardized to ensure consistent formats and unified naming conventions for shared concepts (e.g., "bridge", "harbor", "ship"). Finally, Mini-InternVL (Gao et al., 2024) is sampled at a low rate to preserve general vision–language understanding without overfitting to generic imagery. This curated data corpus enables BabelRS to learn modality-agnostic, task-relevant representations without requiring spatially aligned multi-modal data.

### 5.2. Finetuning Dataset

We evaluate heterogeneous multi-modal object detection on the SOI-Det (Li et al., 2026) benchmark, which combines datasets from three heterogeneous sensing modalities: SAR, optical, and infrared. The benchmark includes SARDet-100K (Li et al., 2024c) for SAR imagery, DOTA-v1.0 (Xia et al., 2018) for optical aerial images, and DroneVehicle (Sun et al., 2022) for infrared vehicle detection. Together, these datasets cover a diverse set of object categories, imaging resolutions, and annotation formats, including both horizontal and oriented bounding boxes.

### 5.3. Implementation Details

We conduct language-pivoted pretraining using a modern, well-engineered vision–language model (VLM) framework. Training a VLM from scratch typically requires multiple stages and substantial computational resources. To reduce this cost, we initialize BabelRS from InternVL-2.5 1B (Chen et al., 2024a), which employs a variant of ViT-Large (Chen et al., 2024b) visual backbone and a Qwen2 (Bai et al., 2023) language model. For the set of feature maps $\mathcal{V}$ in Layerwise Visual-Semantic Annealing, we leverage the 3rd, 9th, 18th and last layer of the ViT-Large, as suggested in (Li et al., 2025b; Bolya et al., 2025). During downstream tasks, the ViT is wrapped into a standard ViT-adapter (Chen et al., 2022) as the backbone.

All pretraining experiments are performed on 8× NVIDIA

*Table 1.* Composition of the language-pivoted pretraining dataset used in BabelRS, including dataset size, sampling rate, and task type (VQA: visual question answering; VG: visual grounding; CLS: classification).

| Dataset | Size | Sample Rate | Tasks |
|---|---|---|---|
| Mini-InternVL (2024) | 1394k | 0.01 | VQA |
| RSVQA (2020) | 100k | 1 | VQA |
| FIT_RS (2024) | 100k | 0.2 | VQA |
| GeoChat (2024) | 64k | 1 | VG |
| VRSBench (2024b) | 38k | 1 | VG |
| DIOR-RSVG (2023) | 27k | 1 | VG |
| VHM (2025) | 223k | 1 | VQA |
| LevirCC (2022) | 50k | 0.2 | Caption |
| GAIA (2025) | 33k | 1 | Caption |
| Million-AID (2021) | 920k | 0.03 | Caption,CLS |
| MMRS-1M (2024a) | 52k | 1 | VQA |
| SARLang (2025) | 1126k | 0.6 | VQA |

A40 (48 GB) GPUs with a global batch size of 128 and learning rate of 2e-5. Fine-tuning is conducted using the same hardware configuration and a unified training pipeline. Standard data augmentation, dataset sampling strategies, and normalization protocols follow prior work (Li et al., 2026). We use the AdamW optimizer with a learning rate of 5e-5, weight decay of 0.05, and a per-GPU batch size of 4. Evaluation follows standard object detection protocols. We report AP at IoU = 0.5 (AP@50), global mean AP averaged over IoU thresholds from 0.5 to 0.95 (mAP), and the proposed Harmonic Modality mAP (H-mAP) to assess balanced performance across modalities.

### 5.4. Main Results

### 5.5. Compare with SOTAs

Table 2 reports heterogeneous multi-modal object detection performance on the SOI-Det benchmark. The compared methods predominantly focus on *fine-tuning stage optimization*, introducing various alignment or regularization mechanisms during downstream training. In contrast, our proposed **BabelRS** emphasizes *pretraining stage optimization* through early, language-pivoted semantic alignment, and employs only a simple joint training strategy during fine-tuning. As shown in Table 2, BabelRS achieves the best performance across all evaluation metrics. Notably, the improvements are consistent across all three sensing modalities. In particular, BabelRS exhibits substantial gains on SAR and infrared datasets, where general-purpose visual pretraining typically struggles due to limited modality coverage.

These results demonstrate that BabelRS learns a balanced and modality-agnostic representation that generalizes effectively across heterogeneous sensing domains. The strong

*Table 2.* Heterogeneous multi-modal object detection performance on the SOI-Det benchmark. All compared methods focu on fine-tuning stage optimization, whereas BabelRS emphasizes pretraining stage optimization and uses a simple joint fine-tuning strategy. BabelRS achieves the best performance.

| Method | Test on | AP@50 | **mAP** | **H-mAP** |
|---|---|---|---|---|
| *Fine-tuning stage optimization* | | | | |
| Simple Joint Training (2020) | Overall | 77.56 | 47.05 | |
| | SARDet-100K | 84.11 | 53.46 | 47.57 |
| | DOTA | 76.37 | 45.18 | |
| | DroneVehicle | 73.28 | 44.99 | |
| DA (2019) | Overall | 79.76 | 48.37 | |
| | SARDet-100K | 84.93 | 53.86 | 49.23 |
| | DOTA | 78.47 | 46.23 | |
| | DroneVehicle | 77.43 | 48.21 | |
| UniDet (2022) | Overall | 79.55 | 48.47 | |
| | SARDet-100K | 84.70 | 53.81 | 49.24 |
| | DOTA | 78.28 | 46.49 | |
| | DroneVehicle | 77.17 | 47.99 | |
| Uncertainty loss (2018) | Overall | 79.99 | 48.79 | |
| | SARDet-100K | 84.81 | 53.43 | 49.57 |
| | DOTA | 78.73 | 46.94 | |
| | DroneVehicle | 77.96 | 48.78 | |
| SM3Det (2026) | Overall | 80.68 | 50.20 | |
| | SARDet-100K | 89.94 | 60.64 | 51.31 |
| | DOTA | 77.88 | 46.47 | |
| | DroneVehicle | 77.99 | 48.87 | |
| *Pretraining stage optimization* | | | | |
| **BabelRS** (ours) | Overall | **81.32** | **51.57** | |
| | SARDet-100K | 91.70 | 63.30 | **53.02** |
| | DOTA | 77.73 | 46.96 | |
| | DroneVehicle | 79.63 | 51.32 | |

*Table 3.* Comparison of different pretraining strategies on SOI-Det. All models use a ViT-Large backbone and identical fine-tuning protocols. BabelRS achieves the best performance across all metrics.

| Method | Test on | AP@50 | **mAP** | H-mAP |
|---|---|---|---|---|
| *Late Alignment (General Backbones)* | | | | |
| CLIP (2021) | Overall | 65.21 | 36.12 | |
| | SARDet-100K | 72.70 | 42.00 | 37.12 |
| | DOTA | 62.67 | 33.56 | |
| | DroneVehicle | 63.83 | 36.74 | |
| MAE (2022) | Overall | 72.36 | 42.84 | |
| | SARDet-100K | 70.64 | 39.48 | 42.54 |
| | DOTA | 73.35 | 43.43 | |
| | DroneVehicle | 71.46 | 45.11 | |
| BEiT (2021) | Overall | 76.50 | 44.59 | |
| | SARDet-100K | 81.90 | 50.10 | 44.94 |
| | DOTA | 76.39 | 43.14 | |
| | DroneVehicle | 70.34 | 42.35 | |
| BEiTv2 (2022) | Overall | 77.63 | 44.67 | |
| | SARDet-100K | 78.50 | 46.70 | 45.35 |
| | DOTA | 78.01 | 43.36 | |
| | DroneVehicle | 75.48 | 46.14 | |
| DINOv2 (2023) | Overall | 74.79 | 45.02 | |
| | SARDet-100K | 72.74 | 41.38 | 44.34 |
| | DOTA | 76.53 | 46.23 | |
| | DroneVehicle | 72.04 | 45.74 | |
| *Late Alignment (Remote Sensing Backbones)* | | | | |
| RemoteCLIP (2024) | Overall | 66.37 | 37.17 | |
| | SARDet-100K | 73.90 | 43.30 | 38.30 |
| | DOTA | 63.57 | 34.36 | |
| | DroneVehicle | 65.74 | 38.27 | |
| SatMAE (2023) | Overall | 74.49 | 42.51 | |
| | SARDet-100K | 79.70 | 47.90 | 43.03 |
| | DOTA | 73.86 | 40.84 | |
| | DroneVehicle | 70.12 | 41.07 | |
| ScaleMAE (2023) | Overall | 74.09 | 42.52 | |
| | SARDet-100K | 78.80 | 46.50 | 43.15 |
| | DOTA | 73.22 | 41.01 | |
| | DroneVehicle | 71.07 | 42.30 | |
| *Early Alignment* | | | | |
| **BabelRS** (ours) | Overall | **81.32** | **51.57** | |
| | SARDet-100K | 91.70 | 63.30 | **53.02** |
| | DOTA | 77.73 | 46.96 | |
| | DroneVehicle | 79.63 | 51.32 | |

H-mAP score further confirms that the gains are not driven by a single remote sensing modality but reflect robustness across modalities.

## 5.6. Comparison with Other Pretraining Methods

We further compare BabelRS against a wide range of pre-training methods in Table 3. For a fair comparison, all methods employ a ViT-Large backbone and are fine-tuned using the same *simple joint training* protocol. CLIP-style pretraining methods, including CLIP (Radford et al., 2021) and RemoteCLIP (Liu et al., 2024), perform poorly on dense object detection tasks. This is largely due to their reliance on global semantic alignment at the final layer, which is insufficient for spatially detection. General-purpose self-supervised backbones such as MAE (He et al., 2022), BEiT (Bao et al., 2021), BEiTv2 (Peng et al., 2022), and DINOv2 (Oquab et al., 2023) also struggle to generalize across heterogeneous remote sensing modalities, particularly when exposed to SAR and infrared data during fine-tuning. Even remote

sensing–specific pretraining methods, SatMAE (Cong et al., 2022) and ScaleMAE (Reed et al., 2023), fail to adequately address heterogeneous multi-modal detection. These methods lack explicit cross-modal alignment priors during pretraining, forcing modality alignment to occur implicitly during fine-tuning. Such *late alignment* significantly complicates optimization and limits generalization.

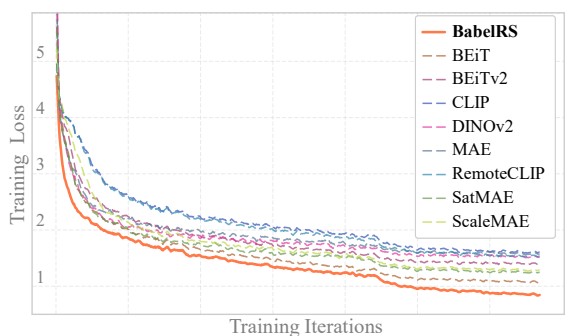

*Figure 4.* Training loss curves under identical finetuning protocols. Late-alignment methods exhibit slow convergence, while BabelRS starts from a lower initial loss and converges smoothly.

*Table 4.* Detection performance under AMP training on SOI-Det. Several late-alignment methods suffer from numerical instability, whereas BabelRS remains stable and achieves strong performance.

| Pretrain Method | AP@50 | mAP | H-mAP |
|---|---|---|---|
| MAE (2022) | NaN | NaN | NaN |
| BEiTv2 (2022) | NaN | NaN | NaN |
| DINOv2 (2023) | NaN | NaN | NaN |
| ScaleMAE (2023) | NaN | NaN | NaN |
| CLIP (2021) | 64.58 | 35.62 | 36.68 |
| RemoteCLIP (2024) | 65.97 | 36.82 | 37.72 |
| SatMAE (2022) | 70.49 | 39.45 | 40.00 |
| BEiT (2021) | 75.74 | 43.82 | 44.35 |
| BabelRS (Ours) | **79.13** | **50.17** | **51.52** |

In contrast, BabelRS consistently and significantly outperforms all late-alignment approaches across all metrics. These results clearly demonstrate the effectiveness of early, language-pivoted semantic alignment and highlight the limitations of relying solely on fine-tuning stage alignment for heterogeneous multi-modal detection.

### 5.7. Optimization Stability and Training Dynamics

Beyond accuracy, we analyze optimization stability, which is a central motivation of this work. Figure 4 compares fine-tuning loss curves under identical optimization settings. Late-alignment methods exhibit unstable behavior, including slow convergence and, in some cases, divergence. In contrast, BabelRS starts from a significantly lower initial loss and maintains smooth and stable convergence throughout training. This empirical evidence supports our core claim: decoupling modality alignment from task learning via early semantic alignment leads to a better-conditioned optimization landscape and substantially reduces gradient interference across modalities.

This stability advantage becomes even more pronounced under Automatic Mixed Precision (AMP) training. As shown in Table 4, several late-alignment methods including MAE,

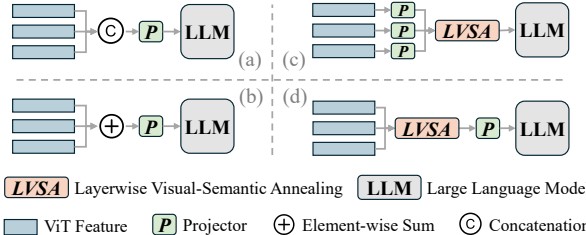

*Figure 5.* Comparison of feature merge strategies: (a) feature concatenation, (b) element-wise summation, (c) per-layer projectors with LVSA, and (d) the proposed LVSA-based merge with a shared projector.

BEiTv2, DINOv2, and ScaleMAE, suffer from severe numerical instability and fail to converge, resulting in gradient explosion. Even methods that remain trainable under AMP experience substantial performance degradation.

In contrast, BabelRS remains fully stable under AMP training and achieves strong performance across all metrics. This robustness is particularly important for large-scale training, where AMP is widely adopted to reduce memory consumption and accelerate optimization.

Figure 7 in the Appendix further illustrates this behavior. Several late-alignment methods exhibit sharp gradient norm spikes and erratic loss trajectories during finetuning, indicative of gradient conflicts between modality alignment and detection objectives. BabelRS maintains well-controlled gradient norms and smooth loss decay, empirically confirming that early language-pivoted alignment substantially improves optimization robustness under aggressive training regimes.

### 5.8. Compare to other merge strategies

We evaluate different strategies for merging intermediate ViT features in Figure 5 and Table 5. The baseline follows the vanilla InternVL design, where only the final-layer feature is passed to the projector. Naïve feature concatenation (Configuration (a)) or element-wise summation (Configuration (b)) of intermediate layers leads to limited improvements, while assigning independent projectors to each layer (Configuration (c)) introduces additional complexity and instability. In contrast, our proposed LVSA-based merge strategy (Configuration (d)), which gradually fuses multiscale features followed by a shared projector, achieves the best performance across all metrics. These results confirm the effectiveness of controlled, progressive multi-scale integration.

### 5.9. Pretraining Steps and Annealing Schedule

Figure 6 analyzes the effect of pretraining duration and the LVSA annealing schedule. As shown on the left, detection

*Table 5.* Detection performance of different intermediate feature merge strategies corresponding to Figure 5.

| Configuration | mAP | H-mAP |
|---|---|---|
| Baseline | 49.33 | 50.67 |
| (a) | 50.25 | 51.60 |
| (b) | 50.31 | 51.55 |
| (c) | 49.88 | 50.92 |
| (d) Ours | 51.57 | 53.02 |

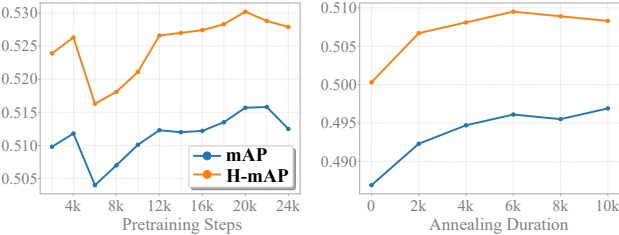

*Figure 6.* Effect of pretraining duration (left) and LVSA annealing schedule (right) on SOI-Det dataset performance.

*Table 6.* Pretraining efficiency of BabelRS v.s. other SOTA remote sensing pretraining methods. Time is estimated for pretraining on 8× A40 GPUs.

| Methods | Hours |
|---|---|
| RVSA (Wang et al., 2023b) | 250 |
| Scale-MAE (Reed et al., 2023) | 60 |
| RemoteCLIP (Liu et al., 2024) | 100 |
| SkySense (Guo et al., 2023) | 400 |
| **BabelRS** | **20** |

performance initially drops around 6k pretraining steps. We attribute this to the early involvement of intermediate-layer features before they are sufficiently optimized. As pretraining continues, both mAP and H-mAP improve steadily, saturating at approximately 20k steps. A slight degradation beyond this point suggests mild overfitting.

We further study the impact of the annealing parameter $\tau$. For efficiency, all models are fine-tuned using AMP. As shown on the right of Figure 6, both mAP and H-mAP consistently improve as $\tau$ increases from 0 to 6k steps, demonstrating that longer and smoother language-pivoted alignment yields stronger cross-modal representations. Performance peaks at $\tau = 6k$, indicating that a moderate and gradual incorporation of multi-scale features is crucial. Larger values of $\tau$ do not provide further gains, suggesting that overly slow annealing is unnecessary. Based on these observations, we set $\tau = 6k$ in all experiments.

### 5.10. Training Efficiency

Beyond detection performance, an important practical advantage of BabelRS lies in its substantially improved pretraining efficiency. As shown in Table 6, BabelRS requires only 20 GPU hours on an 8×A40 setup, significantly reducing computational cost compared with prior remote sensing pretraining frameworks. BabelRS demonstrates that stable cross-modal alignment does not require prohibitively expensive pretraining. Instead, early semantic alignment through language supervision offers a computationally lightweight yet highly effective alternative, making large-scale heterogeneous multi-modal remote sensing detection substantially more accessible for practical deployment and future research.

## 6. Conclusion

This paper introduces BabelRS, a language-pivoted pretraining framework for heterogeneous multi-modal remote sensing object detection. By explicitly decoupling modality alignment from task-specific learning, BabelRS addresses the optimization instability inherent in late-alignment paradigms. Concept-Shared Instruction Aligning enables implicit cross-modal alignment without requiring spatially paired data, while Layerwise Visual-Semantic Annealing bridges the granularity gap between language semantics and dense detection features. Extensive experiments demonstrate the superior effectiveness of BabelRS.

## Acknowledgments

This work is supported by National Natural Science Foundation of China (62361166670, U24A20330, 62576177, 62206134, 62225604), Shenzhen Science and Technology Program (QNXMB202507010908101002, JCYJ20250604184027034, JCYJ20240813114237048), Guangdong Basic and Applied Basic Research Foundation (2026A1515011435), the Tianjin Key Laboratory of Visual Computing and Intelligent Perception (VCIP) and the Fundamental Research Funds for the Central Universities (070-63263247, 070-63253222, 070-63253217). Computation is supported by "Science and Technology Yongjiang 2035" key technology breakthrough plan project (2025Z053), Chinese government-guided local science and technology development fund projects (scientific and technological achievement transfer and transformation projects) (254Z0102G).

## Impact Statement

This paper presents work whose goal is to advance the field of Machine Learning. There are many potential societal consequences of our work, none which we feel must be specifically highlighted here.

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

# A. Theoretical Analysis of Early- and Late-alignment

In this section, we provide a theoretical analysis to support the claim that *late alignment* strategies for heterogeneous multi-modal remote sensing detection induce unstable optimization dynamics, and that separating modality alignment from task learning yields improved stability. Our analysis focuses on gradient interference, loss geometry, and conditioning effects in joint optimization.

## A.1. Problem Formulation

Let $\mathcal{M} = \{m_1, \ldots, m_K\}$ denote a set of modalities, each associated with a data distribution $P_m(x, y)$ over images $x$ and detection labels $y$. A unified detector consists of a shared visual backbone $E_\theta$ parameterized by $\theta$, followed by a detection head $D_\psi$.

Late-alignment methods optimize the joint objective

$$\min_{\theta, \psi} \sum_{m \in \mathcal{M}} \mathbb{E}_{(x,y) \sim P_m} \left[ \mathcal{L}_{\text{det}}(D_\psi(E_\theta(x)), y) + \lambda \mathcal{L}_{\text{align}}(E_\theta(x), m) \right], \tag{13}$$

where $\mathcal{L}_{\text{align}}$ enforces cross-modal feature consistency during fine-tuning.

In contrast, BabelRS decomposes learning into two stages:

1. **Pretraining (alignment only):**
$$\min_{\theta} \sum_{m \in \mathcal{M}} \mathbb{E}_{(x,q,r) \sim P_m} \mathcal{L}_{\text{lang}}(E_\theta(x), q, r), \tag{14}$$

2. **Fine-tuning (task only):**
$$\min_{\theta, \psi} \sum_{m \in \mathcal{M}} \mathbb{E}_{(x,y) \sim P_m} \mathcal{L}_{\text{det}}(D_\psi(E_\theta(x)), y). \tag{15}$$

We now analyze why the joint objective in Eq. (13) is intrinsically ill-conditioned.

## A.2. Gradient Interference Across Modalities

Let $g_m(\theta)$ denote the gradient of the detection loss for modality $m$:

$$g_m(\theta) = \nabla_\theta \mathbb{E}_{(x,y) \sim P_m} \mathcal{L}_{\text{det}}(D_\psi(E_\theta(x)), y).$$

In heterogeneous remote sensing scenarios, modalities arise from fundamentally different physical imaging mechanisms (e.g., SAR scattering vs. optical reflectance). As a result, their optimal representations occupy distinct subspaces. This induces large angular discrepancies between gradients:

$$\cos(g_{m_i}, g_{m_j}) = \frac{\langle g_{m_i}, g_{m_j} \rangle}{\|g_{m_i}\| \|g_{m_j}\|} \ll 0, \quad i \neq j.$$

**Proposition 1 (Gradient Conflict).** If there exists a pair $(m_i, m_j)$ such that $\langle g_{m_i}, g_{m_j} \rangle < 0$, then the variance of the stochastic gradient estimator grows with model capacity, leading to unstable updates.

The joint gradient $g = \sum_m g_m$ has norm

$$\|g\|^2 = \sum_m \|g_m\|^2 + \sum_{i \neq j} \langle g_{m_i}, g_{m_j} \rangle.$$

Negative cross terms increase variance and amplify sensitivity to minibatch composition. As backbone dimensionality grows (e.g., ViT-Large), these effects scale superlinearly, resulting in gradient explosion or numerical instability.

Late alignment exacerbates this issue by introducing an additional alignment gradient $g_{\text{align}}$, whose optimal direction differs from modality-specific detection gradients.

## A.3. Ill-Conditioned Joint Loss Geometry

Consider the Hessian of the joint objective in Eq. (13):

$$H = \nabla_\theta^2 \left( \mathcal{L}_{\text{det}} + \lambda \mathcal{L}_{\text{align}} \right) = H_{\text{det}} + \lambda H_{\text{align}}.$$

In heterogeneous settings, $H_{\text{det}}$ is highly anisotropic, as different modalities induce curvature along incompatible directions. Meanwhile, $H_{\text{align}}$ enforces feature collapse across modalities, introducing sharp curvature along alignment dimensions.

**Proposition 2 (Condition Number Explosion).** If the principal eigenspaces of $H_{\text{det}}$ and $H_{\text{align}}$ are misaligned, then the condition number

$$\kappa(H) = \frac{\lambda_{\max}(H)}{\lambda_{\min}(H)}$$

grows with $\lambda$ and model depth, leading to unstable optimization under first-order methods.

This explains empirical observations of NaNs and divergence when scaling late-alignment models or introducing dynamic routing components such as Mixture-of-Experts.

## A.4. Benefits of Early Language-Pivoted Alignment

BabelRS avoids joint optimization instability by performing modality alignment *implicitly* through language-conditioned pretraining, rather than via explicit feature matching. Unlike contrastive or regression-based alignment objectives, BabelRS does not enforce direct geometric proximity between visual and linguistic embeddings.

Let $E_\theta$ denote the shared visual encoder and $\Phi$ a pretrained large language model (LLM). Given an image $x$ from any modality $m$, the encoder produces a sequence of visual tokens $Z = E_\theta(x)$, which are concatenated with textual instruction tokens $Q$ and fed into the LLM. The model is trained to generate a response sequence $R$ by minimizing a causal language modeling loss:

$$\mathcal{L}_{\text{lang}} = -\log p_\Phi(R \mid Z, Q). \tag{16}$$

Crucially, no explicit constraint is imposed on the distance between visual and textual representations. Instead, alignment arises from the requirement that visual tokens from different modalities must induce equivalent conditional distributions over language outputs for the same semantic instruction.

**Implicit Semantic Equivalence via Conditional Generation.**    Consider two modalities $m_i$ and $m_j$ observing semantically equivalent scenes. Let $Z_i = E_\theta(x^i)$ and $Z_j = E_\theta(x^j)$ be their visual token representations. Language-pivoted alignment enforces:

$$p_\Phi(R \mid Z_i, Q) \approx p_\Phi(R \mid Z_j, Q), \tag{17}$$

for shared instructions $Q$ and responses $R$.

This induces semantic equivalence under the LLM's decision boundary, rather than metric alignment in feature space. As a result, modality-specific visual features are encouraged to encode information that is functionally interchangeable with respect to semantic reasoning, while still preserving modality-dependent low-level structure.

**Optimization Implications.**    This form of alignment has three important optimization consequences:

1. **No feature collapse:** Since alignment is enforced at the level of conditional likelihood rather than embedding distance, modality-specific representations are not forced into a single narrow subspace.

2. **Smooth supervision signal:** Gradients are mediated through the LLM's pretrained language manifold, which exhibits a smooth loss geometry due to large-scale instruction tuning.

3. **Decoupled curvature sources:** Alignment gradients arise solely from the language modeling objective and are applied prior to detection fine-tuning, eliminating curvature interference between semantic alignment and dense prediction objectives.

**Gradient Coherence After Language-Pivoted Pretraining.** After pretraining, the shared encoder produces representations that are semantically normalized across modalities. Consequently, downstream detection gradients become more coherent.

**Proposition 3 (Improved Gradient Alignment).** Let $g_m(\theta)$ denote the detection gradient for modality $m$ after language-pivoted pretraining. Then,

$$\mathbb{E}\big[\langle g_{m_i}, g_{m_j} \rangle\big] \geq 0, \tag{18}$$

for most modality pairs $(m_i, m_j)$, up to higher-order residual terms.

By enforcing semantic equivalence through conditional generation, early language-pivoted alignment reduces representational discrepancy prior to task learning. This mitigates destructive gradient interference and yields a better-conditioned optimization landscape for downstream detection.

## B. AMP Training Loss and Gradient Norm Trajectories

To further analyze the optimization stability of heterogeneous multi-modal detectors, we examine training dynamics under Automatic Mixed Precision (AMP), which is a standard setting for large-scale model training. AMP reduces numerical precision during forward and backward passes and therefore serves as a stress test for optimization robustness.

Figure 7 compares the training loss and gradient norm trajectories of late-alignment baselines and the proposed BabelRS under identical AMP configurations. Some late-alignment methods (DINOv2, MAE, BEiTv2 and ScaleMAE) exhibit pronounced gradient norm spikes accompanied by unstable loss behavior, and in several cases diverge with NaN values. This instability arises from the tight coupling of cross-modal feature alignment and task-specific detection optimization, which induces severe gradient conflicts when learning from heterogeneous modalities.

In contrast, BabelRS maintains smooth loss curves and well-controlled gradient norms throughout training. This stability is a direct consequence of early, language-pivoted semantic alignment during pretraining, which decouples modality alignment from downstream detection optimization. As a result, fine-tuning operates on semantically aligned feature distributions, yielding a better-conditioned optimization landscape that remains robust under reduced numerical precision.

These results demonstrate that the improved AMP stability of BabelRS is not an implementation artifact, but rather reflects a fundamental advantage of early semantic alignment for large-scale heterogeneous multi-modal remote sensing detection.

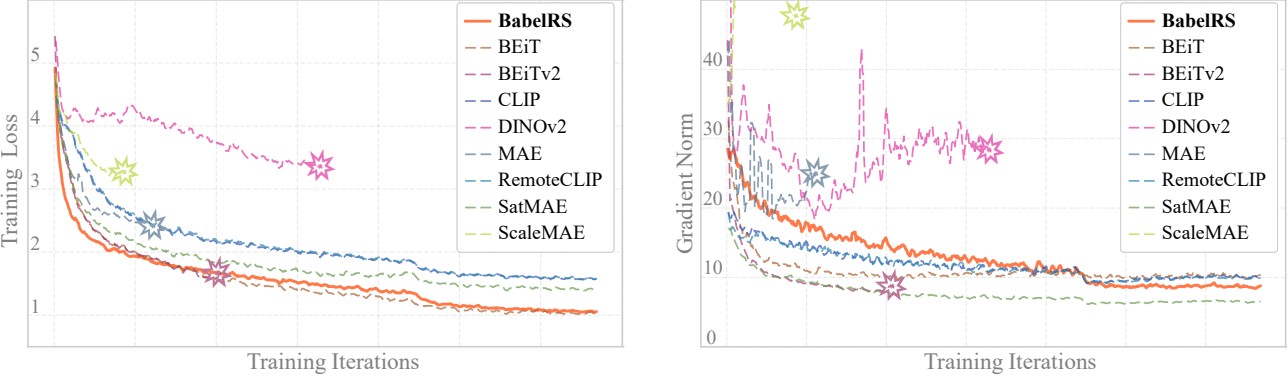

*Figure 7.* Training loss and gradient norm trajectories under AMP training. Late-alignment methods exhibit sharp gradient norm spikes and unstable loss behavior, whereas BabelRS maintains controlled gradients and smooth convergence, demonstrating superior numerical stability.

## C. Examples of Visual–Language Instruction-Following Data

To illustrate the instruction-following data used during pretraining, we provide representative examples spanning different remote sensing modalities. Each sample consists of an input image, a natural-language instruction, and the corresponding model-generated response during pretraining.

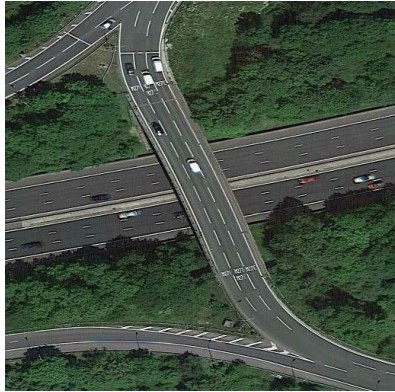

**Subdataset:** Million-AID (Long et al., 2021)

**Modality:** Optical (RGB)

**Task Type:** Visual Question Answering

**Instruction:** Upon careful examination of the image, determine which category(s) it best fits from the provided list: industrial land|public service land|unutilized land|agriculture land|residential land|transportation land|commercial land|water area.

**Response:** The image is a top-down view of a four-way road intersection. The image is in color and has a high resolution. The road surface is asphalt and is in good condition. There is a small amount of traffic on the road. There are trees and shrubs on either side of the road. There is a building in the background. Based on the above reasoning, it is: "transportation land".

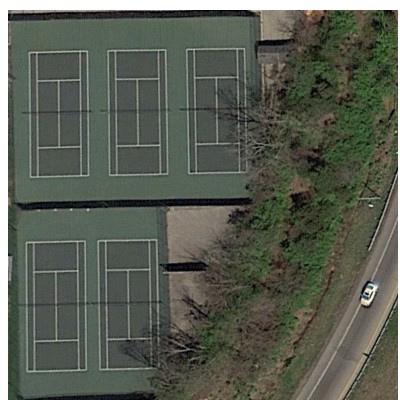

**Subdataset:** DIOR_RSVG (Zhan et al., 2023)

**Modality:** Optical (RGB)

**Task Type:** Visual Grounding

**Instruction:** Please provide the bounding box coordinate of the region this sentence describes:⟨ref⟩ The tennis court is on the lower right of the tennis court on the upper left⟨/ref⟩.

**Response:** ⟨ref⟩The tennis court is on the lower right of the tennis court on the upper left⟨/ref⟩⟨box⟩[[231, 587, 388, 922]]⟨/box⟩.

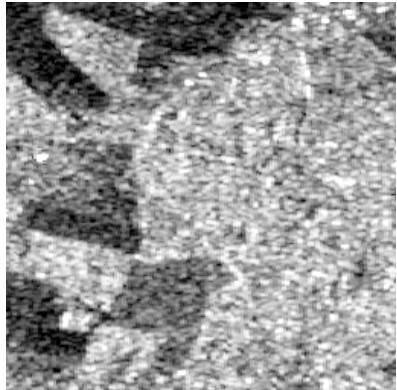

**Subdataset:** SAR_Sentinel (2025)

**Modality:** Synthetic Aperture Radar (SAR)

**Task Type:** Classification

**Instruction:** Select the correct category for this image: agriculture land|grass land|barren land|urban. Use one word or a short phrase to describe the category.

**Response:** Agriculture land.

---

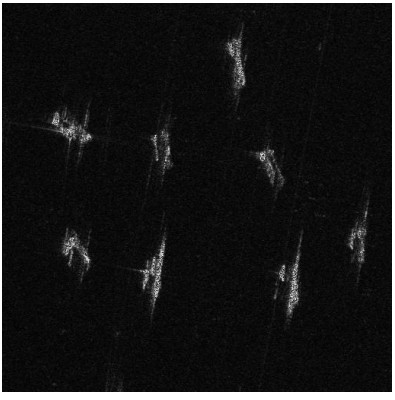

**Subdataset:** SARLang (Wei et al., 2025) and SARDet-100K (Li et al., 2024c)

**Modality:** Synthetic Aperture Radar (SAR)

**Task Type:** Visual Refering

**Instruction:** What is the category visible in the area ⟨box⟩[[294, 46, 35, 69]]⟨/box⟩? The classes are: tank, car, aircraft, harbor, bridge, ship.

**Response:** A ship.

---

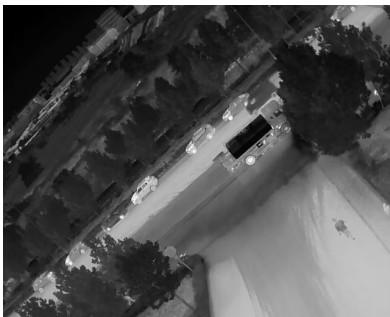

**Subdataset:** MMRS-1M (Zhang et al., 2024a)

**Modality:** Infra-Red (IR)

**Task Type:** Visual Grounding and Counting

**Instruction:** Detect all objects shown in this infrared remote image and describe using horizontal bounding box. If an object category appears more than once, output the quantity only.

**Response:** There are ten objects shown in the image, including seven cars, a truck ⟨box⟩[[550,340,750,530]]⟨/box⟩, and two people.

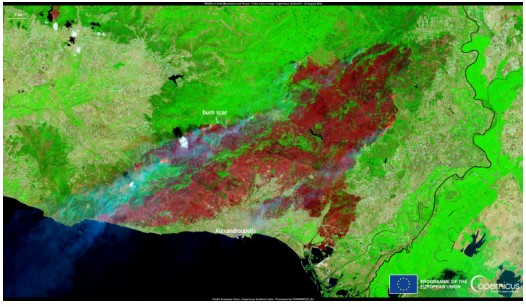

**Subdataset:** GAIA (Zavras et al., 2025)

**Modality:** Multi-Spectral

**Task Type:** Visual Grounding and Counting

**Instruction:** What is the main idea of this image?

**Response:** A vast burn scar in East Macedonia and Thrace, Greece, captured by the Sentinel-2 satellite. The high spatial resolution of the MultiSpectral Instrument (MSI) reveals the extensive damage caused by wildfires, with a distinct contrast between the charred areas and surrounding green vegetation.

## D. Limitation and Future Work

Despite its strong empirical performance, BabelRS has several limitations that warrant further investigation. The effectiveness of language-pivoted cross-modal alignment inherently depends on the coverage, granularity, and correctness of language supervision. Although instruction-following data provides rich and flexible semantic guidance, certain fine-grained, rare, or highly domain-specific targets (particularly in specialized SAR or thermal imaging scenarios) may not be adequately described by natural language alone. This limitation may restrict alignment quality for subtle structural patterns or sensor-specific physical phenomena that lack clear linguistic counterparts.

Moreover, the current framework focuses primarily on static image-level object detection. Although LVSA enables multi-scale feature integration, BabelRS does not explicitly model temporal dependencies or long-range cross-frame

correlations, which are critical for many real-world remote sensing applications involving time-series observations or multi-pass acquisitions.

Future work will address these limitations along several directions. One promising avenue is to enhance language supervision through adaptive or hierarchical prompting strategies, potentially incorporating expert-defined vocabularies or physics-aware descriptions to better cover specialized remote sensing semantics. Another direction is the development of hybrid backbone architectures that combine a shared, language-aligned semantic core with lightweight modality-aware adaptation modules. Such designs may preserve cross-modal consistency while better capturing sensor-specific characteristics.

In addition, extending language-pivoted alignment beyond object detection to support heterogeneous multi-modal and multi-task learning, including semantic segmentation, instance segmentation, and video-based detection, represents a natural and impactful progression. Integrating temporal modeling and multi-sensor time-series data into the BabelRS framework may further improve robustness and generalization, paving the way toward unified remote sensing foundation models capable of handling diverse modalities, tasks, and spatiotemporal scales.

