# OpenReview forum: "Unifying Heterogeneous Multi-Modal Remote Sensing Detection Via Language-Pivoted Pretraining"
_ICML.cc/2026/Conference — ICML 2026 regular_

### Official Review · Reviewer_XQLN · 2026-03-12

**Soundness:** 3
**Presentation:** 2
**Significance:** 3
**Originality:** 3
**Overall Recommendation:** 4
**Confidence:** 4

**Summary:**

This paper addresses the remote sensing object detection problem. The research context refers to leveraging information encoded in images obtained from different sensors (referred to as multi-modal in the paper). The main novelty of the paper relies on a language-based approach to bridge and “semantically” align information provided by different sensors. The goal is performing object detection without the need of having (spatially) paired samples. The proposed method comprises two main modules: CSIA, which is charge of aligning different modalities to the same linguistic concepts; and LVSIA, which aggregates visual features for semantic guidance.

**Compliance With Llm Reviewing Policy:**

Affirmed.

**Final Justification:**

The provided clarifications and additional experiments improved the presentation and scientific relevance of the paper.

**Key Questions For Authors:**

Other remarks:

Other comments and issues:
1.	Given the location where Figure 2 is referred to, it is unclear what the figure is meant to convey. Which baselines would the method illustrated refer to?
2.	“Existing unified detection frameworks for spatially heterogeneous remote sensing data typically entangle modality alignment with task learning.”: This sentence lacks references.
3.	The description of the causal language modelling is shallow and unclear.
4.	It is unclear why only RGB and SAR images alignment scenarios are considered.
5.	The rationale for using an annealing formulation for combining feature representation is unclear. Simply assigning learnable wights to different layers could lead to the same effects.
6.	The description of the task-specific fine-tuning is unclear. The meaning of a task specific loss is not described sufficiently clear. The meaning of $Loss_n$ is unclear.
7.	It is not clear in Section 3.4 what the authors mean by modality-specific failures in the context of an object detection problem.
8.	It is unclear what the authors mean by dataset standardization. Are there images with multiple objects (categories)? Some examples to illustrate data pre-processing would contribute to the readability of the paper.
9.	Why only one dataset is used in experiments? Some statistics (object/category) distribution per sensor type should have been provided.
10.	What is motivation for the use of InternVL-2 1B? What would be the performance of the method if other VLM was used?
11.	The validation protocol is unclear, especially for the pretraining phase. Which component of Figure 3 is being trained? Which trained model is used to compute the detection results?
12.	Some elaboration regarding the relevance of using Automatic Mixed Precision (AMP) is missing in the paper.
13.	Figure 5 describes experiments related to the comparison with different merge strategies. The different considered approaches are not defined clearly.
14.	How would the baselines perform on the examples discussed in the appendix?
15.	The paper lacks discussion on the computational complexity of the proposed method.
16.	The problem formulation math notation presented in appendix lacks rigor.
17.	Appendix B: which baseline considered in the experiments is based on a self-supervision paradigm?

**Limitations:**

Limitations are addressed in the appendix. Overall, the provided discussion is sound.

**Strengths And Weaknesses:**

The target problem is not formally defined. Only in Section 3.1, the training pre-requisites for the method, image-text pairs, is explicitly mentioned. Also, motivational aspects concerning literature gaps are not fully convincing (“empirical observations”). The provided discussion is tailored around numerical failures and instability (gradient explosion). Even though those are relevant problems in the object detection task based on sensor fusion, their connection with a solution based on a language-pivoted pretraining is not clearly established. Another point of lack of consistency refers to the originally aspects discussed in the related work section. There, the training instability problems are not addressed any longer and other literature gaps are discussed. This mismatch weakens the scientific take-home message of the paper.

The proposed method is not described in a sound way (specific comments are provided below). The paper lacks clarity regarding the formulation of the pre-training step and in the object detection phase. Figure 3, for example, which is expected to provide an overview of the proposed method is very confusing. What is the data flow in the pre-training stage? As the method is described, the study reproducibility is compromised.

Validation is not fully convincing. The selection of baselines and the training protocol are not described clearly. Comparisons lacks results of recently proposed methods.

---

> ### Author Rebuttal · Authors · 2026-03-26
>
> **For Weaknesses:**
>
> We respectfully disagree with some statements. The target setting is already formally introduced in Sec. 3.1 and the appendix. We also proper cited the related work [SM3Det], which our paper builds upon. We will revise the presentation accordingly, but we do not think it is accurate to say the problem is not defined.
>
> The central claim of the paper is precisely that late alignment entangles cross-modal reconciliation with dense detection optimization, whereas language-pivoted pretraining decouples these objectives. This argument is already present in the introduction, method, appendix analysis, and AMP experiments.
>
> We do not see a fundamental mismatch between the gaps discussed in the introduction and the related work section, but we agree that the positioning can be made more consistent. The paper is intended to contrast BabelRS against two limitations of prior work: dependence on paired multi-sensor data and entanglement of alignment with downstream detection optimization. We will make these two axes explicit throughout both the introduction and the related work sections.
>
> We agree that Figure 3 can be improved, but we think "not described in a sound way" is too strong. The intended data flow is already specified in Sec. 3. But we will redraw the figure and rewrite Sec. 3.1-3.3 to make this flow easier to follow, but the underlying training pipeline is already described in the manuscript.
>
> We will release the code and dataset upon acceptance to ensure reproducibility.
>
> We follow standard practices in Sec. 4.3 "Standard data augmentation, ... work (Li et al.,2026).", Please refer to this paper for more details. We will summarize these key components directly in the paper to improve clarity, rather than relying on external references.
> Also see Reviewer iyTz Q2 & Reviewer oU8W Weaknesses 6.
>
> **Key Questions:**
>
> 1. We respectfully believe the figure and caption are consistent with the main text. However, we will revise the surrounding explanation to make the intended comparison more explicit.
>
> 2. Thanks for pointing. The paper [SM3Det] should be referred to again here. We will add the missing references in the revision.
>
> 3. We respectfully disagree with this. The objective is explicitly defined in Sec. 3.1, including the image/instruction/response formulation and the causal language modelling loss over response tokens.
> We agree that a brief clarification will improve accessibility and will add a concise explanation without introducing unnecessary background.
>
> 4. We respectfully clarify that the paper consistently considers RGB, SAR, and infrared modalities across all sections (abstract, method, datasets, and experiments). Please review our paper carefully, as it represents the culmination of our team's long-term efforts. Please.
>
> 5. We disagree that simple learnable weighting is equivalent in practice. Static learnable weights still expose the model to full multi-scale fusion from the start of pretraining, which is exactly what LVSA is designed to avoid. The point of annealing is not just expressive power, but controlled optimization. This is also empirically supported in our study: the more flexible alternatives do not match the proposed design, and the per-layer-projector variant is, in fact, worse despite being more complex. We will make this optimization rationale clearer, but we believe the current ablation already supports the need for annealing.
>
> 6. `Loss_n` denotes the "task-specific losses n" as clearly stated in the paper. It is the loss function associated with the sampled branch/dataset during downstream training (as SM3Det does). We will clarify this notation.
>
> 7. We refer to scenarios where a unified detector performs well on dominant modalities but underperforms on others. Global averages can obscure this imbalance, motivating H-mAP.
>
> 8. Dataset standardization: unifying instruction formats, harmonizing semantic labels (e.g., “ship”, “harbor”), normalizing bounding box formats. The dataset includes multi-object, multi-category scenes.
>
> 9. See Reviewer iyTz Q4. Statistics can be found in the referenced paper [SM3Det], where the dataset is proposed.
>
> 10. We clarify that our implementation is based on InternVL-2.5 1B, not InternVL-2 1B.
> We choose this model because it is one of the few open-source VLMs with a ViT-Large visual encoder, making it well-suited for integration with standard detection backbones. In addition, the related work [ViTP] is also built on InternVL-2.5 1B, where its feasibility has already been validated.
>
> 11. See Reviewer eCQ2 Q9.
>
> 12. See main paper Line 404-407.
>
> 13. We respectfully disagree with this. We think Figure 5 and Section 4.8 explain clearly enough for readers who have relevant background knowledge.
>
> 14. Examples in the appendix are from the curated dataset for pretraining, not even related to the baseline models.
>
> 15. See Reviewer oU8W Weaknesses 4
>
> 16. See Reviewer iyTz Q5
>
> 17. MIM methods and image-image contrastive learning methods.

---

> > ### Author Rebuttal · Reviewer_XQLN · 2026-04-03
> >
> > I appreciate the authors' efforts to provide clarifications and insights. The improved presentation helps ensure that the paper’s scientific contributions and reproducibility are clearly conveyed. Also, the additional experiments address previous concerns about the lack of comparisons with recent baselines. I will change my scores accordingly.

---

### Official Review · Reviewer_oU8W · 2026-03-13

**Soundness:** 3
**Presentation:** 3
**Significance:** 3
**Originality:** 2
**Overall Recommendation:** 5
**Confidence:** 4

**Summary:**

This paper focuses on the problem of heterogeneous multimodal object detection, relaxes the requirements for input modalities, and trains a single detector that can generalize across modalities, thereby enabling independent inference on RGB, SAR, or infrared images. Meanwhile, this paper identifies several limitations in existing works: current methods adopt a late alignment strategy, where cross-modal alignment and task optimization are performed simultaneously during the fine-tuning stage for downstream detection tasks, leading to optimization conflicts and training instability.To address this issue, this paper propose BabelRS, a language-pivoted pre-training framework. The method achieves cross-modal alignment through two key modules: Concept-Shared Instruction Aligning (CSIA), Layerwise Visual-Semantic Annealing (LVSA).

**Compliance With Llm Reviewing Policy:**

Affirmed.

**Final Justification:**

The rebuttal has fully addressed my concerns.

**Key Questions For Authors:**

I have no questions.

**Limitations:**

The paper has not discussed the limitations.

**Strengths And Weaknesses:**

Strengths

1.	The paper proposes a relatively novel training approach and identifies the shortcomings of previous training methods. It shifts from late alignment to early, language-pivoted alignment.

2.	The semantic pivot method proposed in this paper is innovative to a certain extent. By using language as the pivot for cross-modal semantic alignment, it avoids the dependence on spatially paired multimodal datasets.

3.	The experimental results are sufficient and comprehensive.The authors analyzed the stability of different pre-training strategies and various backbones under AMP training.
Weaknesses

1.	The analysis of methodological complexity in this paper is insufficiently clear. BabelRS relies on initialization from large-scale vision-language models and employs multiple datasets for pre-training. The paper should fully discuss the computational cost of the model and provide corresponding analysis.

2.	The ablation study in the paper needs to be further strengthened to thoroughly discuss the individual contributions of CSIA and LVSA.

3.	The paper should compare with more recently proposed methods in Table 3.

---

> ### Author Rebuttal · Authors · 2026-03-26
>
> Thank you for the positive assessment and for recognizing the novelty of early language-pivoted alignment, the strength of the empirical study, and the AMP analysis. We address your concerns below.
>
> Weaknesses 4:
> We agree that the cost discussion should be more explicit. On 8 A40 GPUs, 6k pretraining steps of BabelRS take about 16 hours. In contrast, under our implementation setting, ScaleMAE takes about 60 hours and RemoteCLIP about 100 hours. More importantly, BabelRS concentrates the extra cost in a single pretraining stage; downstream detection fine-tuning does not require an additional alignment branch. We will add a clearer complexity discussion in the revision.
>
> Weaknesses 5:
> We agree that this can be presented more clearly. The current paper already provides evidence for both components: the pretraining-step study in Fig. 6 left subfigure reflects the benefit of CSIA. The diagram starts with step 0, which means there is no CSIA that takes part in. While the annealing study in Fig. 6 right subfigure (also starts from 0, meaning no LVSA that takes part in) and the merge-strategy comparison in Fig. 5 / Table 5 isolate the role of LVSA. In particular, the proposed LVSA design performs best among all merge variants. We will make this decomposition more explicit in the revision.
>
> Weaknesses 6:
> Actually, we also tested newer pretrained models such as AIMv2(2025) and SigLIP2(2025), but in our setting both became numerically unstable very early in training, even under FP32, so we do not currently have reliable results to report. We will mention this more clearly in the revision and expand the comparison where stable and fair transfer settings are available.
>
> About Limitation:
> We discussed the limitation in the appendix, as acknowledged by other reviewers. But we will further discuss it in the revision according to the reviewer iyTz's suggestion.

---

> > ### Author Rebuttal · Reviewer_oU8W · 2026-04-01
> >
> > I have no other questions and I will keep the weak accept rating as no more recent methods are compared and it seems that the author can not finish it in several days.
> >
> > I appreciate the efforts that the authors made, so I would like to raise the score to accept.

---

> > > ### Author Response · Authors · 2026-04-02
> > >
> > > Thank you for your continued assessment. In response to this concern, we made an additional effort during the rebuttal period. We gathered as many computational resources as we could and found a better implementation of newer pretrained models. Specifically, we were able to establish stable and fair fine-tuning settings for AIMv2 (2025), SigLIP2 (2025), and Perception Encoder (Meta, 2025).
> > >
> > > To ensure a fair comparison, we carefully tuned the optimization hyperparameters for each model. The fine-tuning learning rates are set to 2e-5, 1.5e-5, and 1.5e-5, respectively, with layer-wise learning rate decay factors of 0.75, 0.8, and 0.7.
> > >
> > > We report below the newly obtained results on the SOI-Det benchmark, which can be considered as an extension to Table 3:
> > >
> > > | Model                 | AP@50 | mAP  | H-mAP |
> > > |----------------------|-------|------|-------|
> > > | AIMv2 (2025)         | 68.45 | 37.89 | 39.04 |
> > > | SigLIP2 (2025)       | 70.56 | 39.52 | 41.02 |
> > > | Perception Encoder (2025) | 76.53 | 43.59 | 44.05 |
> > > | BabelRS (Ours) | 81.32 | 51.57 | 53.02 |
> > >
> > > These results provide a more complete and up-to-date comparison with recent advances. We will incorporate these experiments into the revised manuscript and further analyze their implications in relation to our proposed method.
> > >
> > > We hope this additional effort helps address your concerns. If you find these new results satisfactory, we would greatly appreciate it if you could reconsider your 'Acknowledgement' option selection and overall rating.

---

### Official Review · Reviewer_iyTz · 2026-03-13

**Soundness:** 2
**Presentation:** 3
**Significance:** 2
**Originality:** 3
**Overall Recommendation:** 3
**Confidence:** 3

**Summary:**

This paper studies heterogeneous multi-modal remote sensing object detection in a practically important setting where training data come from RGB, SAR, and infrared sensors and are not spatially paired across modalities. The paper argues that existing unified detectors mostly follow a late-alignment paradigm, where modality alignment and downstream detection optimization are entangled during fine-tuning, leading to optimization conflicts, unstable training, and limited generalization. To address this, the authors propose BabelRS, which consists of CSIA (Concept-Shared Instruction Aligning) and LVSA (Layerwise Visual-Semantic Annealing). CSIA uses language as a semantic pivot to align different modalities into a shared semantic space, while LVSA progressively incorporates intermediate visual features to mitigate the granularity mismatch between language supervision and dense detection. The paper reports gains over SM3Det and multiple pretraining baselines on SOI-Det, and further emphasizes improved AMP stability.
From the problem perspective, this is a real and nontrivial setting: not standard paired multimodal fusion, but heterogeneous, unpaired, unified detection. This direction is meaningful and aligns with a broader trend in multimodal learning from task-time fusion toward earlier semantic alignment during pretraining. ImageBind showed that not all modality pairs need to be directly paired to obtain a shared embedding space; LanguageBind made language the center of multimodal semantic alignment; and UNIALIGN further explored scalable unified alignment across arbitrary modalities. Accordingly, the main contribution here is not the abstract idea of language-pivoted alignment itself, but rather its adaptation to heterogeneous remote sensing detection, together with an attempt to address dense-prediction-specific multi-scale needs.
Overall, I find the paper well motivated, empirically nontrivial, and directionally meaningful, but I am not yet convinced that the current version clears the bar for ICML main-track acceptance. My main reasons are: first, the originality relative to the closest language-centered / unpaired multimodal alignment literature is moderate rather than strong; second, the empirical evidence is too narrow and misses the closest comparison points; third, the “theoretical analysis” reads more like mechanism-motivated explanation than rigorous theory; and fourth, there is a potentially serious evaluation contamination concern on the SAR side that must be clarified. Under ICML’s emphasis on soundness and significance, these issues materially affect the overall recommendation.

**Compliance With Llm Reviewing Policy:**

Affirmed.

**Key Questions For Authors:**

1. Please provide a precise decontamination protocol between the pretraining corpus and the SOI-Det evaluation data, especially for the SAR branch. Since SARLang is used during pretraining and public documentation states that SARLANG-1M includes SARDet-100K imagery as one of its sources, it is important to clarify whether any validation/test images, near-duplicates, cropped variants, or transformed versions could have appeared in pretraining. A convincing clarification here would materially affect my confidence in the reported SAR-side gains.
2. Why are the closest conceptual baselines for language-centered or unified multimodal alignment not included in the empirical comparison? Since the core claim is that early language-pivoted alignment is preferable to late alignment for heterogeneous detection, it would be important to understand whether the gains are specific to BabelRS or would already arise from a more general early-alignment recipe.
3. Can the authors better disentangle the gains from (i) language-pivoted alignment itself, (ii) instruction-style pretraining, (iii) LVSA, and (iv) the particular initialization strategy? At present, several design choices are introduced together, which makes causal attribution difficult.
4. The paper is framed at the level of a pretraining paradigm, but the evidence is concentrated on a single benchmark. Do the authors have additional evidence, even in preliminary form, that the same principle transfers to another heterogeneous detection benchmark or to another dense-prediction task? This would substantially strengthen the significance claim.

**Limitations:**

No. The paper acknowledges some technical limitations, but the discussion of broader risks is not yet adequate. ICML’s research ethics guidelines explicitly state that when risks are directly associated with the proposed methods, application, or data usage, authors are expected to make a reasonable effort to identify those risks and discuss them thoughtfully, ideally with mitigation strategies where feasible. Here, the work targets heterogeneous remote-sensing detection across SAR, optical, and infrared imagery, and the paper’s own data/examples involve categories and concepts such as tank, aircraft, harbor, ship, and infrared vehicle detection, which makes dual-use concerns difficult to dismiss as merely speculative. I would encourage the authors to expand the limitations/impact discussion to address possible misuse in surveillance, military, or other high-risk settings, and to more clearly describe intended boundaries of deployment.

**Strengths And Weaknesses:**

1. This paper addresses a meaningful and technically nontrivial problem: heterogeneous multi-modal remote sensing detection without requiring spatially paired data across RGB, SAR, and infrared sensors. The motivation is clear and timely. The proposed framework, BabelRS, is built around a coherent hypothesis: prior late-alignment approaches entangle modality alignment with downstream detection optimization, whereas language-pivoted pretraining may decouple these objectives and improve both optimization stability and generalization. In that sense, the paper is well grounded in an important practical setting and is aligned with a broader trend in multimodal learning toward earlier semantic alignment.

2. From a soundness perspective, the paper has several strengths. The empirical section is more substantial than that of many application-driven submissions: beyond the main results on SOI-Det, the authors include comparisons to multiple pretraining strategies, AMP analysis, and ablations on feature aggregation and annealing. This gives the impression of a carefully developed empirical study rather than a single-table result. At the same time, the soundness case is not fully closed. Most of the evidence is concentrated on a single benchmark, which feels somewhat narrow given the broader framing of the work as a pretraining paradigm. In addition, the “theoretical analysis” appears more like an explanatory argument for why early alignment may be preferable than a rigorous theory contribution in its own right. A further concern is the possible overlap between SARLang-based pretraining data and SARDet-100K imagery, since public information indicates that SARLANG-1M includes SARDet-100K as a data source. This does not establish contamination, but it does create a nontrivial evaluation concern that should be explicitly clarified.

3. In terms of presentation, the paper is generally clear, well structured, and easy to follow. The roles of the two main components are understandable: CSIA provides language-centered semantic alignment, while LVSA is introduced to recover finer-grained features needed for dense detection. The overall narrative from motivation to method to experiments is coherent. The main weakness here is not readability, but positioning. Although the paper cites the relevant conceptual neighbors, it does not yet distinguish its contribution sharply enough from the closest frontier literature on multimodal alignment. In particular, the manuscript would benefit from a more explicit articulation of how BabelRS differs from prior multimodal alignment approaches and why those differences matter specifically for heterogeneous remote-sensing detection.

4. Regarding significance, I view the paper positively, especially within the remote-sensing and multimodal perception community. Removing the assumption of spatially paired cross-sensor data is practically important, and the paper has the potential to influence future work on unified remote-sensing foundation models. However, the current submission demonstrates significance more strongly at the domain level than at the level of the broader ICML audience. Since the evaluation remains limited to one benchmark and one task family, the paper does not yet fully establish that the proposed pretraining principle will generalize beyond this specific setting.

5. On originality, I would assess the work as moderately original rather than strongly original. Its originality lies less in introducing a fundamentally new multimodal principle and more in adapting and recombining recent ideas in a thoughtful way for a difficult setting: unpaired heterogeneous remote sensing with dense detection. That is a legitimate form of originality under ICML’s criteria. At the same time, the high-level idea of aligning heterogeneous modalities through a shared semantic interface is already well represented in recent multimodal work, so the novelty of BabelRS seems to come primarily from its task-specific formulation and LVSA design rather than from a wholly new learning paradigm. This makes the contribution real, but somewhat bounded.

6. Overall, I see this as a technically serious and well-motivated paper with meaningful strengths in problem formulation, methodological coherence, and empirical effort. Its main weaknesses are the limited breadth of evidence, insufficiently sharp positioning against the closest prior work, and the need for a more explicit clarification of possible data overlap on the SAR side. In its current form, I find the paper promising and credible, but not yet fully convincing at the level expected for a strong ICML main-track acceptance.

---

> ### Author Rebuttal · Authors · 2026-03-26
>
> We greatly appreciate your positive feedback as well as constructive suggestions for improvement. We hope that our rebuttal adequately addresses your concerns.
>
> **Q1: Potential Data Contamination**
>
> We apologize that we accidentally omitted such details in the submission.
>
> In the final pretraining corpus used for all reported results, we removed benchmark-derived samples that overlap with downstream evaluation data. Concretely, we filtered SARDet-100K-derived content from SARLang, and DOTA-derived content from GeoChat, VRSBench, and VHM during corpus construction.
>
> But in case you are interested, actually, during the very early stages of our project, we did forget to filter out certain data, which led to a data leak. Interestingly, unfiltered data did not improve performance and slightly degraded results, possibly due to instruction-style pretraining biasing representations. While we did not include such results in the paper, we agree this is an interesting direction for future study.
>
> **Q2: Positioning Against Prior Multimodal Alignment Literature and Missing Strong Baselines for Early Multimodal Alignment**
>
> BabelRS is not claiming the general idea of language-centered alignment as a new multimodal principle. We study heterogeneous remote-sensing detection without spatially paired data, and show that moving alignment to pretraining is effective in this setting. In other words, the novelty of BabelRS lies primarily in its task-specific formulation for unpaired heterogeneous data.
> In the revision, we will make this distinction clearer along three axes:
>
> (i) prior general multimodal alignment frameworks are not designed for heterogeneous unpaired multimodal data (as already mentioned at the end of the Related Work section), but we do.
>
> (ii) several remote-sensing multimodal methods rely on paired cross-sensor data, whereas our setting uses disjoint multimodal image-text corpora.
>
> (iii) existing language-alignment approaches typically align semantics at the final layer, while our target task requires spatially informative multi-scale features, which motivates LVSA.
>
> This also explains the baseline choice. The closest conceptual neighbors are important references, but they are not drop-in heterogeneous detection baselines for SOI-Det, nor do they provide directly comparable remote-sensing transfer setups under the same detector/backbone configuration. Therefore, our current experiments focus on two controlled comparison axes:
>
> (i) strong unified detectors that perform alignment during downstream training, and
>
> (ii) strong ViT-Large pretraining strategies transferred under the same downstream detector and fine-tuning protocol.
> We will make this rationale much more explicit in the revision.
>
> **Q3: Insufficient Disentanglement of Design**
>
> To clarify, in BabelRS, language-pivoted alignment is implemented through instruction-style pretraining; they are not two separate components. The current paper already provides evidence for both LVSA and initialization: the right subfigure of Fig. 6 (where annealing duration starts from 0) and the merge-strategy comparison in Fig. 5 / Table 5 isolate the role of LVSA, and the proposed LVSA design performs best among all merge variants. The left subfigure of Fig. 6 reflects the effect of pretraining progress relative to direct initialization from InternVL2.5 1B (step 0). We agree, however, that this decomposition is not stated clearly enough, and we will make it more explicit in the revision.
>
> **Q4: Limited Evaluation Scope**
>
> We agree that additional transfer evidence would strengthen the broader significance claim. At the same time, we would like to emphasize that SOI-Det is not a single homogeneous dataset; it is a large scale benchmark thatcombines SARDet-100K, DOTA, and DroneVehicle. To the best of our knowledge, SOI-Det is currently the first and only public benchmark specifically built for heterogeneous RS detection, so there is no second benchmark available at the moment. We will clarify in the revision.
>
> **Q5: Weakness of Theoretical Justification**
>
> The appendix analysis is intended to provide optimization intuition for why early alignment is beneficial in this setting, not to claim a formal theory contribution on the same footing as the method itself. We will therefore soften the wording in the paper and present this section more explicitly as mechanism-driven analysis that supports the observed training behavior.
>
> **Q6: Limited Broader Impact and Risk Discussion**
>
> Although evaluated on remote sensing, BabelRS is not domain-specific. It addresses heterogeneous, unpaired multimodal learning, which arises in medical imaging, autonomous driving, and other multimodal perception problems. Moreover, we also note that ICML has consistently included works that focus on a single domain-specific task. Therefore, we are confident that our work fits the audience of ICML. We will strengthen the discussion of risks (e.g., dual-use concerns) in the revision.

---

> > ### Author Rebuttal · Reviewer_iyTz · 2026-04-07
> >
> > Thank you for the rebuttal. I appreciate the additional clarifications, and some points are now clearer. However, the main concerns behind my original assessment are still not fully resolved, particularly regarding decontamination, the absence of the closest baseline comparisons, and the limited disentanglement of the proposed design choices. Therefore, I will keep my original score.

---

> > > ### Author Response · Authors · 2026-04-07
> > >
> > > We thank the reviewer for their continued engagement. We believe these remaining concerns stem from a misunderstanding of our paper and rebuttal rather than an unresolved question.
> > >
> > > 1. We respectfully believe that in the initial rebuttal, we have clarified that our implementation ***strictly does not have the issue of decontamination*** as "We filtered SARDet-100K-derived content from SARLang, and DOTA-derived content from GeoChat, VRSBench, and VHM during corpus construction."
> > >
> > > 2. We respectfully believe that in the rebuttal response to Q2, we have ***clarified and explained*** the closest baseline comparisons. If there are ***specific*** additional baselines you feel we should include, we would be more than happy to add them.
> > >
> > > 3. We respectfully believe that both the main paper and our response to Q3 provide a very ***clear and sufficient disentanglement*** of the proposed design choices. Should you feel that a ***specific*** component requires further experimental validation, we would welcome your suggestions.
> > >
> > > Your detailed feedback will help us provide further active clarification and explanation. If our current response and clarification have addressed your concerns, we would be extremely grateful if you would like to reconsider your rating. We sincerely hope you will take the time to carefully understand our work, as your acknowledgement and recommendation rating are critically important to us.

---

### Official Review · Reviewer_eCQ2 · 2026-03-13

**Soundness:** 3
**Presentation:** 3
**Significance:** 2
**Originality:** 2
**Overall Recommendation:** 4
**Confidence:** 4

**Summary:**

Heterogeneous multi-modal remote sensing object detection aims to accurately detect objects from diverse sensors (e.g., RGB, SAR, Infrared).

To address these limitations, the paper proposes BabelRS, a unified language-pivoted pretraining framework that explicitly decouples modality alignment from downstream task learning. BabelRS comprises two key components: Concept-Shared Instruction
Aligning (CSIA) and Layerwise Visual-Semantic Annealing (LVSA).

CSIA aligns each sensor modality to a shared set of linguistic concepts, using language as a semantic pivot to bridge heterogeneous visual representations. LVSA progressively aggregates multi-scale visual features to provide fine-grained semantic guidance.

**Compliance With Llm Reviewing Policy:**

Affirmed.

**Final Justification:**

The authors have answered my remaining questions. These are not fundamental limitations, but should be clarified in the revised version properly.

**Key Questions For Authors:**

Problem Formulation:
1. It's not intuitive why late fusion with different modalities would cause gradient explosion, as shown in Fig. 2. Although it is shown in experimental ablation study, however a technical justification is required.

Methodology:
2. Although objects like “car” in SAR and RGB can be related to the both correspond to the linguistic concept, the objects in SAR would be very hard top relate. Images depicting the same object are mapped to similar features regardless of whether they originate from RGB, SAR, or infrared sensors. However, for other modalities’, this mapping would be extremely noisy.

Concept-Shared Instruction Aligning:  which treats the embedding space of a large language model as a shared semantic reference. Layerwise Visual-Semantic Annealing:  which progressively integrates intermediate ViT representations into the language-aligned space.
It allows the pretraining process preserve the joint calibration of low- and high-level visual features, enabling precise localization while retaining strong semantic guidance from LLMs.

3. The central hypothesis of this work is that while pixel distributions P(xm) and imaging mechanisms vary significantly across modalities, their semantic interpretations can be expressed through shared linguistic concepts. However, it makes a fundamental assumption that the mapping of objects will be consistent like RGB. However, since SAR and even IR, can have objects that are missed, how are these scenarios taken into account ?
4. For each modality, a modality-shared vision encoder EMextracts visual features, which are projected into the input embedding space of Φ. However, these vision encoders may miss objects like SAR, then how is this addressed ? Shouldn’t the Laligned be modality specific ?
Also, its not clear if only the vision encoder is finetuned, or even the LLM is finetuned in an end to end fashion?

5. All the equations have missing equation numbers.
6. Layerwise Visual-Semantic Annealing Mechanism:  I agree that most vision–language models align language only with the final ViT layer, which captures global semantics.  The fused feature representations are a dynamic weighting algorithm, of final and intermediate layers. Although it's interesting way to fuse, it's not clear where is this mechanism used in the training ?








Experimental results:

7. Table 1 highlights the pre-training datasets. Is BabelRS pre-trained on all these datasets ?  Are the methods used for comparison also pre-trained on similar datasets ? Is this a consistent comparison ?
8. If BabelRS uses both pre-training and finetuning datasets, and other SOTA models use only finetuning datasets, then this is not a fair comparison. BabelRS followes InternVL-2.5 1B (Chen et al., 2024a), which employs a variant of ViT-Large (Chen et al., 2024b) pretraining. This is consistent with other SOTA ?
9.Table 3 shows comparison with other ViT-Large pre-training strategies, comparing it with CLIP, DINO etc., However the exact architectural composition for this particular table is not clear. So, how are these ViT-backbones used ?

**Limitations:**

Yes

**Strengths And Weaknesses:**

Soundness: Yes, the submission is sound, with proper experimental ablations.

Presentation: Yes, the paper is well-written and figures are proper.

Significance: Yes, the paper addresses a relevant problem.

Originality: The work is original, with new insights, however, a technical justification is needed, along with experimental ablation results.

---

> ### Author Rebuttal · Authors · 2026-03-26
>
> Thank you for your time and careful review of our paper. We greatly appreciate your positive feedback on the soundness, presentation, significance, and originality of our work, as well as your constructive suggestions for improvement. We have carefully considered all your comments and hope that our rebuttal adequately addresses your concerns.
>
> 1. Our claim is not that any late-fusion design is unstable, but that in heterogeneous RS detection a shared backbone must satisfy two conflicting objectives during fine-tuning: cross-modal alignment and dense detection. Because SAR, RGB, and IR come from very different imaging physics, their gradients can be inconsistent, especially when alignment is enforced only at downstream training time. This is the motivation for the appendix analysis on gradient interference/loss conditioning, and the AMP experiments provide the empirical counterpart. We will connect Fig. 2, the appendix analysis, and the AMP results more clearly in the revision.
>
> 2.&3. BabelRS does **not** assume identical visibility or one-to-one correspondence across RGB, SAR, and IR. Our assumption is weaker: when a modality-specific observation is sufficient to support a concept, language provides a shared semantic anchor even if the appearance differs greatly. Since pretraining is unpaired, we align at the **concept level**, not at the instance level. Thus, objects missed or ambiguous in SAR/IR do not violate the framework; they simply provide weaker or modality-specific supervision through their own image-text pairs. We will make this point explicit in Sec. 3.1.
>
> 4. CSIA is not meant to solve detection during pretraining; it provides a semantically normalized initialization for the visual encoder. Modality-specific localization is then learned during downstream detection fine-tuning. For this reason, we use a **shared semantic space** rather than fully modality-specific aligned spaces, while still keeping modality-specific detection heads so that sensor-specific cues are preserved.  Sorry for the confusion on training details: during pretraining, both the ViT and LLM are trainable; during downstream detection, the LLM is discarded, and only the pretrained ViT is optimized within the detection framework. We will state this explicitly in the revision.
>
> 5. We agree. We will add equation numbers and cross-references in the revision.
>
> 6. LVSA is used during **pretraining**, before projection into the language space. Selected intermediate ViT features are gradually fused with the final-layer representation through the annealing coefficient, so multi-scale localization cues can be introduced smoothly into language-pivoted pretraining. During downstream fine-tuning, the auxiliary alignment objective is removed.
>
> 7&8. Table 3 is designed to evaluate transfer performance under different initialization strategies, rather than enforcing identical upstream pretraining corpora across all methods. All models in Table 3 share the same ViT-Large backbone, detector architecture, and fine-tuning pipeline on SOI-Det, ensuring that the comparison isolates the effect of initialization on downstream performance.
>
> We do not retrain all other methods on exactly the same pretraining dataset for several reasons:
>
> - Objective mismatch: The datasets in Table 1 are constructed for instruction-following pretraining, which is not compatible with many baselines. Re-training them under this objective would fundamentally alter their design.
> - Methodological scope: The curated pretraining data and paradigm are integral components of our method.
> - Computational constraints: Reproducing large-scale pretraining for all baselines is prohibitively expensive and not common practice.
> - Standard practice: Modern vision models are typically evaluated as off-the-shelf pretrained models. Importantly, all baselines in our comparison are also pretrained on large-scale datasets: e.g., CLIP and DINOv2 on large-scale natural images, and SatMAE, ScaleMAE, and RemoteCLIP on large-scale remote sensing imagery.
>
> Therefore, the comparison in Table 3 should be interpreted as a comparison between different pretraining strategies under a unified downstream setting, rather than between pretrained and non-pretrained models. We will revise the paper to explicitly clarify this evaluation protocol and avoid potential misunderstanding.
>
> 9. In Table 3, all compared methods are used as **visual initialization strategies** for the same downstream heterogeneous detector, with the same ViT-Large-sized backbone and the same fine-tuning protocol. ViT-Large is wrapped into a standard ViT-Adapter (Vision Transformer Adapter for Dense Predictions, ICLR 2023) to be used as the backbone. We will clarify this architectural setup in the revision.

---

> > ### Author Rebuttal · Reviewer_eCQ2 · 2026-04-04
> >
> > The rebuttal clarifies several missing details such as clarity regarding CSIA and LVSA. I also understand why it may be an issue to re-run different methods with different pre-training strategies, to compare with the proposed method. However, I feel the paper this needs to clarify questions regarding the methodology, especially (2) and (3) technically. Hence, I will maintain my original recommendation.

---

> > > ### Author Response · Authors · 2026-04-06
> > >
> > > We thank the reviewer for the follow-up. We believe these remaining concerns may stem from a ***misunderstanding*** of our paper rather than an unresolved technical flaw. We would like to restate our core design principles as concretely as possible.
> > >
> > > In short, BabelRS does not assume instance-level correspondence or one-to-one visibility across RGB, SAR, and IR. Our pretraining is explicitly unpaired and concept-level: language serves as the shared semantic pivot only when a modality-specific observation is sufficient to support a concept. Objects that are ambiguous or missing in specific sensors do not violate the methodology, they simply provide modality-specific supervision through their own unique image-text pairs.
> > >
> > > As noted by Reviewers iyTz, oU8W, and XQLN, our use of language as a pivot specifically avoids the dependence on spatially paired data.
> > > - ***Reviewer iyTz*** "This paper addresses a meaningful and technically nontrivial problem: heterogeneous multi-modal remote sensing detection without requiring spatially paired data across RGB, SAR, and infrared sensors. "
> > >
> > > - ***Reviewer oU8W*** "By using language as the pivot for cross-modal semantic alignment, it avoids the dependence on spatially paired multimodal datasets."
> > >
> > > - ***Reviewer XQLN*** "a language-based approach to bridge and “semantically” align information provided by different sensors."
> > > ﻿
> > >
> > > Even though we have  already clearly illustrated in our main paper Section 3.1 and the above rebuttal, to further clarify to make sure you understand this is not our technical flaw but possibly your misunderstanding, we further explain from another angle:
> > >
> > > 1. Probabilistic Alignment vs. Deterministic Mapping:
> > >
> > > Unlike contrastive learning methods that enforce a hard, deterministic distance between visual features and text embeddings, our Concept-Shared Instruction Aligning is probabilistic. Alignment is governed by the causal language modeling loss, $L_{lang} = -\log p_{\Phi}(R | Z, Q)$, see Appendix A.4.
> > > The model is not forced to rigidly map a noisy, ambiguous SAR object to a strict coordinate in the semantic space. Instead, the visual tokens $Z$ induce a conditional probability distribution over the text response $R$. If an object is highly degraded or noisy in SAR, this uncertainty is naturally reflected in a softer probability distribution, preventing the destructive, high-variance gradient updates that plague late-alignment feature matching.
> > >
> > >
> > > 2. Unpaired, Modality-Grounded Supervision
> > >
> > > Because our pretraining is spatially unpaired, we do not force a SAR image to answer a query meant for a clear RGB image.
> > > We utilize modality-specific datasets like SARLang , where the ground-truth text explicitly reflects what is characteristic and visible in SAR imagery. If an object type is systematically unobservable or "missed" due to the physics of SAR or IR, the corresponding image-text pairs in our pretraining corpus simply do not contain descriptions of that object. Therefore, the model is never forced to map an "invisible" feature to a concept, completely avoiding the false cross-modality forcing you rightly pointed out as a risk.
> > >
> > >
> > > 3. Empirical Evidence of Methodological Robustness
> > >
> > > The methodology's ability to handle this noise is empirically validated in our modality-specific results. If mapping SAR and IR to language introduced extreme, destructive noise, we would expect a performance drop in those specific modalities. Instead, BabelRS achieves a massive improvement on SARDet-100K (reaching 91.70 AP@50) and DroneVehicle (reaching 79.63 AP@50), significantly outperforming all baselines. The methodology does not just survive the noisy mapping, it leverages it to improve modality-specific perception.
> > >
> > > We will add related content in Section 3.1. This addition will explicitly detail how the probabilistic nature of the unpaired, modality-grounded instruction data naturally accommodates the high noise and missing objects inherent to SAR and IR. We will also revise the Introduction and Method sections to emphasize these points more explicitly.
> > >
> > >
> > > If there is still a specific technical part of (2) or (3) that remains unclear to you, please ***specify precisely*** which parts you still do not understand and why you believe these issues cannot be resolved within the current time constraints. Your detailed feedback will help us provide further active clarification and explanation.
> > >
> > > If our current response and clarification have addressed your concerns, we would be extremely grateful if you would like to reconsider your rating. We sincerely hope you will take the time to carefully understand our work, as your acknowledgement and recommendation rating are critically important to us.

---

### Decision · Program_Chairs · 2026-04-30

**Decision:**

Accept (regular)

**Comment:**

BabelRS proposes a language-pivoted pre-training that decouples modality alignment from dense detection. The paper is well motivated, and the authors provide solid empirical evidence: significant AP gains on SOI-Det, a clear AMP-stability analysis, and an ablation that isolates CSIA and LVSA.

Reviewers oU8W and XQLN, who initially raised concerns about the motivation and implementation, now find the description satisfactory after the authors' clarifications. Reviewer eCQ2's key questions about the alignment mechanism and the unpaired training have also been addressed in the rebuttal, and has updated his rating. The de-contamination protocol for SAR data, while now described, still warrants a concise, formal statement. The discussion of computational cost and broader impact could be expanded.

Overall, the paper merits acceptance. It is recommended that the authors incorporate the reviewers' suggested revisions in the final version.